# Comparison of the Effects of Laparoscopic and Open Surgery on Postoperative Acute Kidney Injury in Patients with Colorectal Cancer: Propensity Score Analysis

**DOI:** 10.3390/jcm10071438

**Published:** 2021-04-01

**Authors:** Ji Hoon Sim, Sa-Jin Kang, Ji-Yeon Bang, Jun-Gol Song

**Affiliations:** Department of Anesthesiology and Pain Medicine, Asan Medical Center, University of Ulsan College of Medicine, Seoul 05505, Korea; atlassjh@hanmail.net (J.H.S.); sajinkg@naver.com (S.-J.K.); jungol.song@amc.seoul.kr (J.-G.S.)

**Keywords:** laparoscopic surgery, colorectal cancer, acute kidney injury, survival

## Abstract

Postoperative acute kidney injury (AKI) is a serious complication that increases patient morbidity and mortality. However, few studies have evaluated the effect of laparoscopic surgery on postoperative AKI. This study compared the incidence of postoperative AKI between laparoscopic and open surgery in patients with colorectal cancer. This study retrospectively analyzed 3637 patients who underwent colorectal cancer surgery between June 2008 and February 2012. The patients were classified into laparoscopic (*n* = 987) and open (*n* = 2650) surgery groups. We performed multivariable regression analysis to assess the risk factors for AKI and propensity score matching analysis to compare the incidence of AKI between the two groups. We also assessed postoperative intensive care unit (ICU) admission, complications, hospital stay, and 1-year mortality. We observed no significant differences in the incidence of postoperative AKI between the two groups before (8.8% vs. 9.1%, *p* = 0.406) and after (8.8% vs. 7.7%, *p* = 0.406) matching. Laparoscopic surgery was not associated with AKI even after adjusting for intraoperative variables (adjusted odds ratio (OR): 1.17, 95% confidence interval (CI): 0.84–1.62, *p* = 0.355). Body mass index, diabetes mellitus, hypertension, and albumin were risk factors for AKI. ICU admission (0.6% vs. 2.5%, *p* = 0.001), complications (0.2% vs. 1.5%, *p* = 0.002), hospital stay (6.89 days vs. 8.61 days, *p* < 0.001), and 1-year mortality (0.1% vs. 0.9%, *p* = 0.021) were significantly better in the laparoscopic than in the open group. The incidence of postoperative AKI did not differ significantly between laparoscopic and open surgery. However, considering its better surgical outcomes, laparoscopic surgery may be recommended for patients with colorectal cancer.

## 1. Introduction

Acute kidney injury (AKI) is a common postoperative complication that is defined as an abrupt decrease in the kidney function, including decreased urine output, and elevated serum creatinine levels [1]. AKI significantly increases morbidity and mortality, as well as hospital costs in surgical patients [2]. In addition, postoperative AKI increases the risk of developing chronic kidney disease (CKD) and end-stage renal disease (ESRD) [3]. AKI occurs in 5–10% of all hospitalized patients, 4–13.4% of patients who have undergone major abdominal surgery, and up to 60% of intensive care unit (ICU) patients [4,5].

Colorectal cancer (CRC) is the third most common cancer in men and the second most common cancer in women globally, with an annual incidence of about 40 in 100,000 [6]. Surgical treatment plays an important role in colon cancer management and is largely divided into two types: open and laparoscopic. Laparoscopic surgery is well known for resulting in minimal trauma, less bleeding, and more stable hemodynamics in surgical patients [7,8]; moreover, these factors might reduce postoperative acute renal injury [9]. Previous studies have reported lower rates of postoperative AKI for laparoscopic surgery than for open surgery [10,11]. However, few studies have assessed the effect of laparoscopic CRC surgery on postoperative AKI.

Therefore, we compared the incidence of postoperative AKI between open and laparoscopic surgery in patients with CRC. We also assessed postoperative ICU admission and complications, hospital stay, and 1-year mortality.

## 2. Experimental Section

### 2.1. Study Design and Patient Population

This study was approved by the institutional review board (IRB) of Asan Medical Center (protocol number: 2020-1806), and the requirement for written informed consent was waived by the IRB. We retrospectively reviewed all patients diagnosed with CRC based on the International Classification of Diseases, tenth revision (ICD-10), and who underwent open or laparoscopic surgery between June 2008 and February 2012. This study included adult patients aged 18 years and older.

The exclusion criteria were as follows: patients with severe cardiopulmonary or chronic kidney disease before admission; patients who had already received kidney replacement therapy; patients who had received interventions in the urethra, ureter, or kidneys during surgery; patients who underwent emergency surgery; patients with conversion from laparoscopic to laparotomy during surgery; and patients with incomplete data or missing serum creatinine values. 

### 2.2. General Anesthesia and Surgical Technique

All patients, regardless of the approach, were given general anesthesia (open versus laparoscopy). All surgeries were performed by a qualified and experienced surgical team. Conventional and laparoscopic surgeries were performed according to standard protocols [11,12,13]. For laparoscopic surgery, pneumoperitoneum with carbon dioxide at 10–15 mmHg was established.

### 2.3. Clinical Data Collection and Outcome Assessments

All data were obtained from the electronic medical record system. Demographic data, intraoperative variables, and pre- and postoperative laboratory values were collected. Demographic data included age, sex, body mass index (BMI), diabetes mellitus (DM), hypertension (HTN), and American Society of Anesthesiologists (ASA) physical status classification. Intraoperative variables included the surgical method, lowest mean blood pressure (MBP), operative time, infused fluids such as crystalloids or colloids, transfused packed RBC, urine output, inotropes, and diuretics. Pre- and postoperative laboratory values included estimated glomerular filtration rate (eGFR), creatinine, and serum albumin. Postoperative AKI and complications, ICU admission, 1-year mortality rate, and hospital stays were also recorded.

### 2.4. Primary and Secondary Outcomes

The primary outcomes were the results of the comparisons of the incidence of AKI between the two groups and the analysis of the risk factors associated with postoperative AKI. Postoperative AKI was defined according to the Kidney Disease: Improving Global Outcomes (KDIGO) criteria: an increase in serum creatinine (sCr) concentration by >1.5-fold from the preoperative baseline level within 7 days or an absolute increase in creatinine of ≥0.3 mg/dL within 48 h and urine volume <0.5 mL/kg/h for 6 h. Oliguria was defined as a urine output of <0.5 mL/kg/h and was calculated from the intraoperative data [14]. The secondary outcomes were the results of the comparisons of surgical outcomes such as ICU admission, postoperative complications, 1-year mortality rate, and hospital stay between the two groups.

### 2.5. Statistical Analysis

Data were expressed as means ± standard deviations, median (interquartile range), or numbers (proportions), as appropriate. Data variables included in this study were compared between the open surgery and laparoscopic surgery groups using chi-squared or Fisher’s exact tests for categorical variables and independent t- or Mann–Whitney U-tests for continuous variables. We used multiple logistic regression analysis to identify the independent predictors of AKI. All variables with *p* < 0.1 in a univariate analysis were included in the multivariable analysis. We also performed multiple logistic regression analysis to determine the propensity scores using the following seven variables: age, sex, BMI, DM, HTN, ASA, and preoperative laboratory values (albumin, creatinine) (Table 1). Propensity score matching was performed using a greedy algorithm using a caliper of 0.1 standard deviation of the logit of the propensity score. Model calibration was assessed using Hosmer–Lemeshow statistics (χ^2^ = 4.952; df = 18; *p* = 0.999). After performing 1:1 propensity score matching, continuous variables were compared using paired or Wilcoxon signed-rank tests, as appropriate. Categorical variables were compared using McNemar’s or marginal homogeneity tests, as appropriate. After 1:1 propensity score-matching, the final analysis included 987 patients each in the open and laparoscopic surgery groups. In all statistical analyses, *p* < 0.05 was considered significant. All data were analyzed using R (version 3.1.2; R Foundation for Statistical Computing, Vienna, Austria) and IBM SPSS Statistics for Windows, version 22.0 (IBM Corp., Armonk, NY, USA).

## 3. Results

Of the 3843 patients enrolled, 206 were excluded according to the criteria mentioned above. Finally, this study analyzed a total of 3637 patients. Of these, 987 and 2650 underwent laparoscopic and open surgery, respectively (Figure 1).

Table 1 shows the baseline characteristics of the study populations in the unmatched and matched samples. Before matching the preoperative variables, the patients who underwent laparoscopic surgery had a higher BMI (*p* = 0.078), higher baseline albumin (*p* < 0.001), and lower ASA class (*p* < 0.006), with no significant differences in age, sex, DM, HTN, and serum creatinine. After propensity matching, the significant differences that existed before matching between the two groups were eliminated. The intraoperative variables are also listed in Table 1. After matching, the laparoscopic surgery group had a higher lowest MBP (*p* < 0.001) and intraoperative urine output (*p* < 0.001). The operative time for laparoscopic surgery was longer than that for open surgery (*p* < 0.001). The laparoscopic surgery group received lower volumes of total fluid (*p* < 0.001), crystalloid solution (*p* = 0.018), and synthetic colloid solution (*p* = 0.004), as well as fewer RBC transfusions. No patients in the laparoscopic surgery group required the use of inotropes and diuretics, compared to eight and four patients, respectively, in the open surgery group (*p* < 0.001).

### 3.1. Primary Outcomes

The postoperative surgical outcomes between the two groups are shown in Table 2. Before matching, the AKI incidence rates were 8.8% (87/987) in the laparoscopic surgery group and 9.1% (242/2650) in the open surgery group. After matching, the rates were 8.8% (87/987) and 7.7% (76/987), respectively. No significant differences were observed between the two groups both before and after matching (*p* = 0.816 and *p* = 0.406, respectively).

### 3.2. Secondary Outcomes

Both before and after matching, the laparoscopic surgery group demonstrated fewer hospital stays (*p* < 0.001 in the unmatched sample and *p* < 0.001 in the matched sample) and fewer ICU admissions (*p* < 0.001 in the unmatched sample and *p* = 0.001 in the matched sample) (Table 2). Only two patients in the laparoscopic surgery group experienced postoperative complications, and there was a significant difference between the two groups before and after matching (*p* = 0.001 in the unmatched sample and *p* = 0.002 in the matched sample) (Table 2). Regarding 1-year mortality, one patient in the laparoscopic surgery group died, while 25 and 9 patients in the open surgery group died before and after matching, respectively, showing a significant difference between the two groups (*p* = 0.006 in the unmatched sample and *p* = 0.021 in the matched sample) (Table 2).

Laparoscopic surgery was not significantly associated with the incidence of postoperative AKI even after adjusting for other potentially confounding variables (odds ratio (OR) 1.17, 95% confidence interval (CI) 0.84–1.62, *p* = 0.355) (Table 3). However, laparoscopic surgery was significantly associated with reducing the duration of prolonged hospitalization (>14 days) after adjusting for other variables both before and after matching (OR 0.46, 95%CI 0.29–0.74, *p* = 0.001) (Table 3).

In the multivariable analysis, laparoscopic surgery was not significantly associated with the development of postoperative AKI (OR 0.97, 95%CI 0.74–1.27, *p* = 0.829). Conversely, BMI (OR 1.04, 95%CI 1.00–1.09, *p* = 0.035), DM (OR 1.58, 95%CI 1.18–2.13, *p* = 0.002), HTN (OR 1.40, 95%CI 1.06–1.84, *p* = 0.016), and preoperative albumin (OR 0.57, 95% CI 0.44–0.74, *p* < 0.001) were significantly associated with postoperative AKI (Table 4).

## 4. Discussion

The results of our study demonstrated no significant difference in overall postoperative AKI incidence between laparoscopic and open surgeries (8.8% vs. 9.1%) in CRC. The incidence of AKI also did not differ between the two groups after propensity score matching. However, patients who underwent laparoscopic surgery had a shorter hospital stay compared to those who underwent open surgery. The postoperative ICU admission, complications, and 1-year mortality rates were also less for laparoscopic surgery than for open surgery. This finding suggests that laparoscopic surgery may be superior to open surgery in terms of postoperative surgical outcomes in patients with CRC.

Establishing pneumoperitoneum with carbon dioxide is essential for the laparoscopic technique; however, the accompanying increased intra-abdominal pressure (IAP) impaired kidney function in an animal experimental model [12] and caused renal impairment by renal arterial vasoconstriction, ischemia-reperfusion-related hypoxemia, and tubular renal injury during laparoscopic surgery in some human case reports [13,14]. However, recent studies have shown inconsistent results regarding the effects of laparoscopic surgery on postoperative AKI. In laparoscopic cholecystectomy, reductions in eGFR and effective renal plasma flow and urine output have been reported in patients with IAP >12 mmHg [15]. In addition, the urine output is significantly decreased during pneumoperitoneum in laparoscopic adrenalectomy [16]. However, other studies showed significantly lower AKI incidence after laparoscopic surgery compared to that after open surgery [10,11]. Additionally, one study reported no difference in pre- and postoperative urine N-acetyl-beta-D-glucosaminidase levels between the laparoscopic and open surgery groups, suggesting that laparoscopic surgery was not associated with renal tubular injury [17].

The results of the current study are consistent with those of other studies reporting that laparoscopic surgery does not significantly affect renal function or AKI incidence [17,18]. In our study, while the lowest MBP was slightly higher in the laparoscopic group, the difference was not clinically significant, and only eight patients used inotropes during the entire operation. This finding suggests that there is no significant difference in hemodynamic instability between the two groups. Although there may be a temporary decrease in hemodynamics due to a position change and pneumoperitoneum at the beginning of laparoscopic surgery, this phenomenon does not seem to be clinically significant, because no evidence of microscopic damage to the renal tubule has been observed [19]. Therefore, contrary to the claim that laparoscopic surgery can induce hemodynamic instability and AKI, the effects of surgical methods on hemodynamic changes and renal blood flow were small in our study.

In this study, BMI, DM, HTN, and low preoperative albumin levels were independent risk factors for the development of postoperative AKI. A previous study on the association between BMI and AKI post-laparoscopic surgery reported potential confounding factors including DM, HTN, and coronary artery disease [20]. DM and HTN are known risk factors for AKI in patients undergoing general surgery [21]. A possible reason for a high risk of AKI in diabetic and hypertensive patients is the frequent complications of these conditions that can lead to AKI even in the absence of CKD [22].

Low preoperative serum albumin levels are associated with perioperative complications including AKI [23]. Albumin is a potential natural antioxidant that acts as a core extravascular source of reduced sulfhydryl groups, so-called thiols, that scavenge reactive oxygen and nitrogen species; thus, hypoalbuminemia can cause postoperative AKI [24].

The use of synthetic colloids such as hydroxyethyl starch in the ICU and noncardiac surgery has been associated with AKI [25,26]. A prospective randomized controlled trial reported the association between a restrictive fluid regimen and a higher rate of postoperative AKI compared to a liberal fluid regimen [27]. However, other studies reported that synthetic colloids were not associated with postoperative AKI [28] and that excessive fluid administration was associated with a greater risk of AKI because fluid overload may cause renal congestion; reduce renal perfusion and glomerular filtration; and induce atrial dilatation and stretching of blood vessel walls, causing the release of atrial natriuretic peptide (ANP) and damaging the endothelial glycocalyx (EGL), subsequently leading to AKI [29,30]. The results of the multivariable logistic regression in the present study showed that the use of synthetic colloids and total infused fluid volumes were not significantly associated with postoperative AKI. In addition, while hemodilution and transfusion have also been associated with AKI [31], there were very few cases of transfusion in the present study. Thus, the association between administered fluids and postoperative AKI remains controversial, and further well-designed randomized clinical studies are warranted to address this issue.

In the present study, surgical outcomes such as hospital stay, postoperative ICU admission and complications, and 1-year mortality rate were significantly better for laparoscopic surgery than for open surgery, a finding consistent with those previously reported [7,32,33,34,35]. In general, patients who underwent laparoscopic surgery had shorter hospital stays due to the smaller incisions, reduced use of painkillers such as opiates, and early return of bowel functions [33], all of which allowed early patient recovery and return to daily life, increased patient satisfaction, and reduced hospital costs [36]. Additionally, large-scale prospective clinical research has demonstrated that laparoscopic surgery results in fewer postoperative complications [34] and comparable survival rates compared to those for open surgery [37].

This study has several limitations. First, the major limitations of this study are those inherent to a retrospective study. Thus, numerous confounding factors and potential biases related to patient selection and recall may have been present. However, we performed propensity score matching to minimize the effects of these confounding factors. Second, the variables such as cancer stage (tumor size, lymph node invasion, and distant metastasis) and neo-adjuvant treatment were not included in propensity score matching and multivariable analysis. For this reason, some surgical outcomes in our study require careful interpretation. Third, ICU admission, postoperative complications, and 1-year mortality were too few in laparoscopic surgery to be adjusted by intraoperative variables. Therefore, a well-designed prospective study with many cases is required in the future. Fourth, our data were based on the information included in the medical records collected from a single medical center. Thus, the results may have been biased due to similar or homogenous groups, and further multicenter studies on heterogeneous groups are needed.

## 5. Conclusions

In conclusion, although the incidence of postoperative AKI did not differ significantly between the two groups, laparoscopic surgery may be recommended for patients with CRC based on its better outcomes such as hospital stay, postoperative ICU admission and complications, and 1-year mortality.

## Figures and Tables

**Figure 1 jcm-10-01438-f001:**
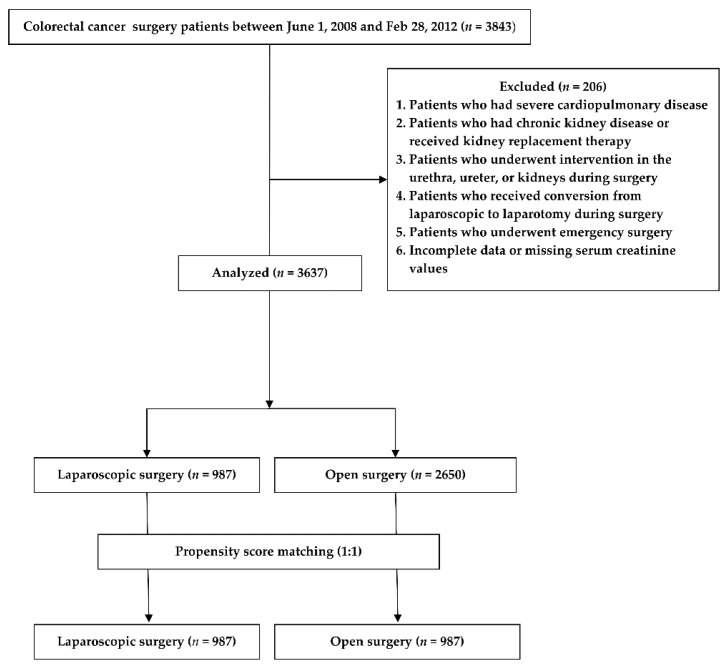
Study flow chart.

**Table 1 jcm-10-01438-t001:** Baseline characteristics of the study populations in the unmatched and matched samples.

	Unmatched Sample				Matched Sample			
	Laparoscopic Surgery (*n* = 987)	Open Surgery (*n* = 2650)	*p*	SMD	Laparoscopic Surgery (*n* = 987)	Open Surgery (*n* = 987)	*p*	SMD
Preoperative variables								
Age; years	59.11 (11.21)	59.92 (11.23)	0.052	0.072	59.11(11.21)	58.99 (11.22)		0.011
Sex; male	601 (60.9)	1631 (61.5)	0.718	0.013	601 (60.9)	604 (61.2)		0.006
BMI; kg·m^−2^	23.99 (3.00)	23.75 (3.14)	0.038	0.078	23.99 (3.00)	24.00 (3.10)		0.004
DM	142 (14.4)	392 (14.8)	0.759	0.011	142 (14.4)	139 (14.1)		0.009
HTN	335 (33.9)	872 (32.9)	0.555	0.022	335 (33.9)	324 (32.8)		0.024
ASA status			0.006	0.127				0.021
ASA 1	254 (25.7)	606 (22.9)			254 (25.7)	256 (25.9)		
ASA 2	722 (73.2)	1974 (74.5)			722 (73.2)	722 (73.2)		
ASA 3	11 (1.1)	70 (2.6)			11 (1.1)	9 (0.9)		
Albumin; g·dL^−1^	3.94 (0.40)	3.81 (0.47)	<0.001	0.294	3.94 (0.40)	3.94 (0.41)		<0.001
Creatinine; mg·dL^−1^	0.80 (0.18)	0.79 (0.18)	0.303	0.038	0.80 (0.18)	0.80 (0.17)		0.018
Intraoperative variables								
Lowest MBP; mm/Hg	72.45 (8.96)	70.30(8.87)	<0.001		72.45 (8.96)	71.13 (8.70)	0.001	
Operation time; min	180.55 (58.25)	172.76 (71.57)	<0.001		180.55 (58.25)	168.81 (68.21)	<0.001	
Total fluids; mL/kg	24.28 (19.40–29.68)	25.76 (20.37–32.87)	<0.001		24.28 (19.40–29.68)	25.32 (19.72–31.98)	0.001	
Crystalloid; mL/kg	18.23 (13.99–23.60)	19.19 (14.34–26.86)	<0.001		18.23 (13.99–23.60)	18.64 (14.00–25.82)	0.018	
Colloid; mL/kg	7.06 (0.00–8.34)	7.35 (0.00–8.92)	<0.001		7.06 (0.00–8.34)	7.17 (0.00–8.79)	0.004	
Colloid use	694 (70.3)	1947 (73.8)	0.06		694 (70.3)	719 (72.8)	0.231	
RBC transfusion	2 (0.20)	118 (4.5)	<0.001		2 (0.20)	36 (3.60)	<0.001	
Urine output; mL/kg/h	1.95 (0.96–3.25)	1.27 (0.68–2.37)	<0.001		1.95 (0.96–3.25)	1.30 (0.71–2.35)	<0.001	
Inotropes	0 (0.0)	15 (0.6)	0.016		0 (0.0)	8 (0.8)	0.008	
Diuretics	0 (0.0)	21 (0.8)	0.002		0 (0.0)	4 (0.4)	0.124	

BMI, body mass index; DM, diabetes mellitus; HTN, hypertension; ASA, American Society of Anesthesiologists classification; SMD, standardized difference; MBP, mean blood pressure; RBC, red blood cells. Values are expressed as the mean (SD), median (interquartile range), or *n* (proportion).

**Table 2 jcm-10-01438-t002:** Surgical outcomes in the unmatched and matched samples.

	Unmatched			Matched		
	Laparoscopic Surgery (*n* = 987)	Open Surgery (*n* = 2650)	*p*-Value	Laparoscopic Surgery (*n* = 987)	Open Surgery (*n* = 987)	*p*-Value
AKI	87 (8.8)	242 (9.1)	0.816	87 (8.8)	76 (7.7)	0.406
ICU admission	6 (0.6)	102 (3.8)	<0.001	6 (0.6)	25 (2.5)	0.001
Postoperative complications	2 (0.2)	35 (1.3)	0.001	2 (0.2)	15 (1.5)	0.002
1-year mortality	1 (0.1)	25 (0.9)	0.006	1 (0.1)	9 (0.9)	0.021
Hospital stay	6.89 (3.31)	8.78 (7.60)	<0.001	6.89 (3.31)	8.61 (7.12)	<0.001

SD, standard deviation; AKI, acute kidney injury; ICU, intensive care unit. Values are expressed as the mean (SD), median (interquartile range), or *n* (proportion).

**Table 3 jcm-10-01438-t003:** AKI incidence and days of hospitalization (>14 days) adjusted for laparoscopic surgery in the matched sample.

	Unmatched				Matched			
	Unadjusted OR (95%CI)	*p*-Value	Adjusted OR * (95%CI)	*p*-Value	Unadjusted OR ** (95%CI)	*p*-Value	Adjusted OR ** (95%CI)	*p*-Value
AKI	0.96 (0.74–1.24)	0.767	0.97 (0.74–1.27)	0.829	1.16 (0.84–1.59)	0.361	1.17 (0.84–1.62)	0.355
Hospital stay (>14 days)	0.38 (0.26–0.56)	<0.001	0.46 (0.30–0.70)	<0.001	0.45 (0.29–0.70)	<0.001	0.46 (0.29–0.74)	0.001

* Adjusted for age, BMI, DM, HTN, operation time, lowest MBP, albumin, total fluids, crystalloid, colloid use, urine output, ASA, and RBC transfusion. ** Adjusted for operation time, lowest MBP, colloid use, and RBC transfusion. AKI, acute kidney injury; SD, standard deviation; OR, odds ratio; CI, confidence interval. Values are expressed as the mean (SD), or *n* (proportion).

**Table 4 jcm-10-01438-t004:** Univariate and multivariate logistic regression analysis of AKI incidence in the unmatched sample.

	Univariate			Multivariate		
	OR	95% CI	*p*-Value	OR	95% CI	*p*-Value
Laparoscopic surgery	0.96	0.74–1.24	0.767	0.97	0.74–1.27	0.829
Age	1.01	1.00–1.02	0.015	1.00	0.99–1.02	0.467
BMI	1.04	1.00–1.08	0.044	1.04	1.00–1.09	0.035
DM	1.76	1.33–2.33	<0.001	1.58	1.18–2.13	0.002
HTN	1.51	1.20–1.91	<0.001	1.40	1.06–1.84	0.016
ASA status						
ASA status 1	1.00	–	–	1.00	–	–
ASA status 2	1.25	0.94–1.66	0.124	0.89	0.64–1.24	0.476
ASA status 3	2.56	1.36–4.79	0.003	1.37	0.68–2.75	0.379
Operation time; min	1.01	0.99–1.02	0.321	1.01	0.99–1.04	0.248
Lowest MBP; mm/Hg	1.00	0.88–1.14	0.997	1.11	0.96–1.27	0.15
Total fluids; mL/kg	1.03	0.96–1.10	0.423	0.90	0.58–1.42	0.666
Crystalloid; mL/kg	1.04	0.96–1.13	0.316	1.09	0.67–1.80	0.722
Synthetic colloid use	0.95	0.74–1.23	0.707	1.03	0.63–1.67	0.918
RBC transfusion	1.23	0.68–2.21	0.488	0.96	0.48–1.91	0.906
Albumin; g·dL^−1^	0.52	0.42–0.66	<0.001	0.57	0.44–0.74	<0.001
Urine output; mL/kg/h	1.04	0.98–1.11	0.218	1.05	0.98–1.12	0.184

AKI, acute kidney injury; SD, standard deviation; OR, odds ratio; CI, confidence interval; BMI, body mass index; DM, diabetes mellitus; HTN, hypertension; ASA, American Society of Anesthesiologists classification; MBP, mean blood pressure; RBC, red blood cells. Values are expressed as the mean (SD), median (interquartile range), or *n* (proportion).

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
