# Peer review of "Comparison of the Effects of Laparoscopic and Open Surgery on Postoperative Acute Kidney Injury in Patients with Colorectal Cancer: Propensity Score Analysis"

_jcm, 2021, doi:10.3390/jcm10071438_

Round 1

Reviewer 1 Report

p3 line 115: definition oliguria < instead of >?

Table 2. the fact that 2.5% of 'open' patients needed ICU postoperatively suggests a more extensive surgery in these patients. Can you say something about the extensiveness of the resection? For rectal cancers do you have information on neo-adjuvant treatment and can you match for location and TNMstage of tumor? I feel your included (baseline) variables are a bit limited to draw these conclusions.

Line 185 Please explain more clearly:  'However, Laparoscopic surgery was significantly associated with hospital stay (>14 days) after adjusting for other variables both before and after matching (OR 0.46, 95%CI 0.29–0.74, p=0.001) (Table 3).'

Author Response

Responses to Reviewer #1

  1. p3 line 115: definition oliguria < instead of >?

Response: We really appreciate your exact comments on our error. We apologize for not describing the symbols correctly. As you pointed out, we have revised and corrected the inequality sign to < instead of > on page 3, line 125-126 as follow, “Oliguria was defined as a urine output of <0.5 mL/kg/hr

  1. Table 2. the fact that 2.5% of 'open' patients needed ICU postoperatively suggests a more extensive surgery in these patients. Can you say something about the extensiveness of the resection? For rectal cancers do you have information on neo-adjuvant treatment and can you match for location and TNM stage of tumor? I feel your included (baseline) variables are a bit limited to draw these conclusions.

Response: We really appreciate your insightful comments. In our study, a total of 25 patients (2.5%) of 'open' patients were admitted to the intensive care unit, of which 7 (0.7%) had intraoperative massive bleeding, 12 (1.2%) had perioperative cardiopulmonary problems, and 6 (0.6%) had other reasons. On the other hand, all 'laparoscopic' patients admitted to the ICU (6, 0.6%) were due to perioperative cardiopulmonary problems. In general, open surgery is preferred over laparoscopic surgery when cancer staging is high or when the surgery is expected to be extensive and difficult. Therefore, open surgery may have worse surgical outcomes than laparoscopic surgery, such as bleeding and transfusion, postoperative complications, and ICU admission. However, in our study, cancer stage (tumor size, lymph node invasion, and distant metastasis) and neo-adjuvant treatment were not considered, which is an obvious limitation of this study. Also, for ICU admission, postoperative complication, and 1-year mortality, propensity matching was performed for preoperative variables, but for intraoperative variables, adjusting was not possible because the number of cases was too small. For this reason, some surgical outcomes in our study require careful interpretation. Therefore, we have further summarized and described the contents of this limitations in the discussion section on page 8, lines 302-307 as follow, Second, the variables such as cancer stage (tumor size, lymph node invasion, and distant metastasis) and neo-adjuvant treatment were not included in propensity score matching and multivariable analysis. For this reason, some surgical outcomes in our study require careful interpretation. Third, ICU admission, postoperative complications, and 1-year mortality were too few in laparoscopic surgery to be adjusted by intraoperative variables. Therefore, a well-designed prospective study with many cases is required in the future.

  1. Line 185 Please explain more clearly: 'However, Laparoscopic surgery was significantly associated with hospital stay (>14 days) after adjusting for other variables both before and after matching (OR 0.46, 95%CI 0.29–0.74, p=0.001) (Table 3).'

Response: Thank you for your kind suggestion. Our expressions can be interpreted somewhat vaguely. Therefore, we have revised the previous sentence to be more clarified. The revised sentence is on page 6, line 202-206 as follow, “However, Laparoscopic surgery was significantly associated with reducing the duration of prolonged hospitalization (>14 days) after adjusting for other variables both before and after matching (OR 0.46, 95%CI 0.29–0.74, p=0.001) (Table 3).”

Reviewer 2 Report

Dear Authors, thank you for allowing me to review your manuscript. I think the paper has some scientific merit, but needs revision:

  1. please check the incidence of CRC - you reported an annual incidence of 1/360,000 (line 40), but unfortunately it is about 40 cases / 100,000 (in South Korea it is 44.5/100,000)
  2. the Materials and Methods section can be reduced, in particular subsection 2.3 is redundant (and a bit confused) and can be deleted to make the paper more readable. Similarly, I am not sure if section 2.2 is essential, it can be easily deleted saying that all patients had general anaesthesia irrespective of the approach (open vs lap).
  3. In line 124 you state that all variables with p>0.1 at univariate analysis were included in the multivariable analysis. I think it should be p<0.1.

Author Response

Responses to Reviewer #2

  1. please check the incidence of CRC - you reported an annual incidence of 1/360,000 (line 40), but unfortunately it is about 40 cases / 100,000 (in South Korea it is 44.5/100,000).

Response: Your comment is right and the authors truly agree with your point. A recent study reported that the incidence of colorectal cancer worldwide is about 30 to 50 / 100,000. Therefore, we have appropriately corrected the incidence of colorectal cancer by citing the literature on page 1, line 40-43.

  1. The Materials and Methods section can be reduced, in particular subsection 2.3 is redundant (and a bit confused) and can be deleted to make the paper more readable. Similarly, I am not sure if section 2.2 is essential, it can be easily deleted saying that all patients had general anaesthesia irrespective of the approach (open vs lap).

Response: It is a valuable comment and we agree to your comment. In response to your comments, we have combined sections 2.2 and 2.3 into one section to make it a bit more readable and concise. This process was demonstrated in Experimental Section on page 2, line 68-105.

  1. In line 124 you state that all variables with p>0.1 at univariate analysis were included in the multivariable analysis. I think it should be p<0.1.

Response: Thank you for your exact point on the wrong inequality sign. we have revised and corrected the inequality sign to < instead of > on page 3, line 137.